# Supervised Detection of Ionospheric Scintillation in Low-Latitude Radio Occultation Measurements

**Vinícius Ludwig-Barbosa** [1,*] , **Thomas Sievert** [1] , **Anders Carlström** [2] , **Mats I. Pettersson** [1] , **Viet T. Vu** [1] and **Joel Rasch** [3]

1   Department of Mathematics and Natural Sciences, Blekinge Institute of Technology,
    371 79 Karlskrona, Sweden; thomas.sievert@bth.se (T.S.); mats.pettersson@bth.se (M.I.P.);
    viet.thuy.vu@bth.se (V.T.V.)
2   RUAG Space AB, 405 15 Gothenburg, Sweden; anders.carlstrom@ruag.com
3   Molflow, 411 01 Gothenburg, Sweden; joel.rasch@molflow.com
*   Correspondence: vinicius.ludwig.barbosa@bth.se

**Abstract:** Global Navigation Satellite System (GNSS) Radio Occultation (RO) has provided high-quality atmospheric data assimilated in Numerical Weather Prediction (NWP) models and climatology studies for more than 20 years. In the satellite–satellite GNSS-RO geometry, the measurements are susceptible to ionospheric scintillation depending on the solar and geomagnetic activity, seasons, geographical location and local time. This study investigates the application of the Support Vector Machine (SVM) algorithm in developing an automatic detection model of F-layer scintillation in GNSS-RO measurements using power spectral density (PSD). The model is intended for future analyses on the influence of space weather and solar activity on RO data products over long time periods. A novel data set of occultations is used to train the SVM algorithm. The data set is composed of events at low latitudes on 15–20 March 2015 (St. Patrick's Day geomagnetic storm, high solar flux) and 14–19 May 2018 (quiet period, low solar flux). A few conditional criteria were first applied to a total of 5340 occultations to define a set of 858 scintillation candidates. Models were trained with scintillation indices and PSDs as training features and were either linear or Gaussian kernel. The investigations also show that besides the intensity PSD, the (excess) phase PSD has a positive contribution in increasing the detection of true positives.

**Keywords:** remote sensing; radio occultation; ionosphere; scintillation; support vector machine

## 1. Introduction

The threats to Global Navigation Satellite System (GNSS) operation caused by the ionosphere are widely known. The ionospheric plasma is composed of free electrons and ions with non-homogeneous distribution. The gradient of electron density can cause rapid fluctuations in the amplitude and phase of signals in the GHz scale, hereafter referred to as scintillations [1,2]. The intensity and occurrence of this phenomenon are extensively influenced by solar and geomagnetic activity, geographical location, seasons and local time [3,4].

GNSS Radio Occultation (RO) is a remote sensing technique, which relies on satellite–satellite trans-ionospheric propagation of GNSS signals, and measurements are performed in L1 (1575.42 MHz) and L2 (1227.42 MHz) bands [5–7]. Given the dispersive characteristic of the ionosphere, a linear combination of both measurements is used to correct the phase shift (Doppler) caused by the ionospheric refractivity [8]. An undesirable residual ionospheric error (RIE) is still observed in atmospheric data of the neutral atmosphere (troposphere and lower stratosphere) after the correction. This issue is most concerning in climatology studies [9,10]. Nevertheless, GNSS-RO operation also benefits research on the ionosphere.

Currently, total electron content (TEC), electron density profiles (EDPs) and, subsequently, the F-layer peak density ($NmF2$), height ($hmF2$) and the respective critical frequency are estimated from the occultations performed by different RO missions [6,11,12]. Hence, these data allow constant monitoring of the conditions in the ionosphere as well as support investigations of different phenomena. TEC measurements performed by the Challenging Minisatellite Payload (CHAMP) mission have been combined with GPS ground-based receivers at high latitudes to create 3-D maps of electron density aided by a tomographic algorithm, narrowing the differences to ionosondes measurements [13]. An assimilation study of simulated RO ionospheric data has shown a reduction of RMS error of TEC, F-layer density and height peak in ionospheric models [14]. Further comparison of Constellation Observing System for Meteorology, Ionosphere and Climate (COSMIC) data to incoherent scatter radars, ionosondes and the International Reference Ionosphere (IRI-2001) model validated the quality of the measurements and substantially increased the amount of RO ionospheric data available [15]. Despite the general assumption of spherical symmetry taken in EDP retrieval [16], the profiles obtained from COSMIC-1 measurements proved to have an accuracy similar to ionosondes and have been applied in solar activity monitoring, especially at low- and mid-latitudes [17]. The current IRI model, version 2016, provides a $hmF2$ model based on CHAMP, GRACE and COSMIC data [18].

RO data also plays an important role in monitoring the effect of space weather events in the ionosphere, for example, geomagnetic storms. An increase in the electron density during the main phase of the "Halloween" storm in November 2003 as well as a decrease in ions upflow during the recovery phase have been observed in RO measurements [19]. During the same storm, the bias in the retrieved refractivity profiles observed in the simulations was three times larger than in the days not affected by such anomaly [20]. During the geomagnetic storm in July 2012, larger areas with strong fluctuations in the southern magnetic polar region and intensified north-south asymmetry in the upper ionosphere were observed in TEC profiles provided by the COSMIC mission [21]. The RO observations by the FengYun-3C (FY3C) satellite showed the variability of the occurrence of scintillations during and after the St. Patrick's Day storm in 2015 according to TEC gradients. The data indicated the absence of scintillation during the main phase of the storm in the African and Eastern Asian sector, whereas the incidence increased significantly in the New Zealand sector. Furthermore, the RO measurements showed an overall lower occurrence of scintillations during the initial recovery phase after the storm [22].

As one of the causes of radio signal scintillation, regions of ionospheric irregularities in the F-layer, known as Equatorial Plasma Bubbles (EPB), have been investigated with RO measurements, adding a limb-sounding perspective to ground-based measurements. During March 2014 (around the equinox and at the maximum of the 24th solar cycle), COSMIC data corroborated ground-based measurements and showed a higher occurrence of EPBs at the magnetic equator after sunset, at about 450 km and descending to 300 km around midnight [23]. A longer period of evaluation, between 2007 and 2017, showed a similar interval of occurrence of EPBs with peak varying depending on the level of solar activity. EPBs occurrences were determined exclusively by the scintillation index ($S_4$). The EPB distribution along the dip equator indicated dependencies of seasons, solar cycle and longitudes [24]. A correlation between strong scintillation and the angle between the occultation ray path and EPBs along magnetic field lines was also verified [25]. Further, RO measurements were combined with ground-based receivers to investigate the occurrence and to characterise scintillation caused by a sporadic E-layer during the daytime in the magnetic dip-equator [26].

Ionospheric scintillation can be alternatively characterised by other statistical parameters, for example, detrended signal phase standard deviation, i.e., $\sigma_\phi$, and the power spectral density (PSD) [2]. Measurements by ground-based receivers had their power spectra extensively used in the investigation of ionospheric irregularities. Power spectra of VHF signals under weak scattering have shown roughly constant power density at low frequencies with a well-defined break point followed by the spectrum decreasing asymptot-

ically to a straight line on log–log scale. Such spectral characteristics added by small-scale ionospheric irregularities in radio signals have been investigated and modelled by different spectral density functions (SDF) [27–32]. The evaluation of signals with different carrier frequencies and their respective intensity PSD showed that low-frequency signals are more prone to strong scintillations causing broadening of the spectrum towards high frequencies, given the larger decorrelation of the signal owing to the multiscattering [33]. Some cases with high $S_4$ index values (strong scattering) have shown different patterns: with monotonic slope throughout the entire frequency range and with the absence of a clear break point; or with two spectral slopes, one shallower slope given by long-scale irregularities and a steeper slope related to short-scale irregularities [34–36].

Phase parameters are most commonly applied in high-latitude measurements, but scintillations in phase also occur at low latitudes and its detection is highly related to coexistent amplitude scintillations [37,38]. Intensity and phase PSD can show the same power spectral slopes and, for some cases under weak scattering, the phase PSD can have a clear minimum around the break point (Fresnel maximum) of the amplitude PSD. This feature has supported the estimation of the scale length of the irregularities in equatorial measurements [39]. Additionally, phase PSD has the advantage of not being affected by Fresnel filtering. Thus, the scale length of irregularities larger than the Fresnel radius can be estimated by combining the distance of the region of irregularities to the receiver and the drift velocity to the phase PSD [40]. In the context of GNSS-RO, studies have considered to a large extent TEC, electron density profiles, $S_4$ index and other complementary geomagnetic indices in analyses and modelling of the ionospheric conditions and regions of irregularities. Few publications have investigated ionospheric scintillation by using intensity and phase spectral analysis and phase scintillation index [41,42].

Machine Learning (ML) can be described as the use of mathematical and statistics tools to create methods able to detect patterns in different sort of data and generalise decisions. The ML algorithms evaluate input features and map them to known outputs (supervised learning) or to an unknown set of outcomes (unsupervised learning). The problems are either of classification, where data are assigned to a finite group of observations sharing common characteristics (qualitative), or regression, in which the outputs of the method are real scalars (quantitative) [43]. Considering the large amount of ionospheric data available, Machine Learning has been applied in the development of detection and prediction models of ionospheric events. The Support Vector Machine (SVM) algorithm [44,45] has been used to develop a model of automatic detection of equatorial amplitude scintillation using $S_4$ and intensity PSD as training features [46]. The model achieved on average a true positive rate (TPR) of 88.75 % with a false positive rate (FPR) of 2.75% when measurements with moderate and strong scintillation were considered in the model training. A similar approach has been used in the detection of phase scintillation at high latitudes using $\sigma_\phi$ and phase PSD as training features [37]. Both models were trained and validated with L-band measurements by ground-based receivers. An SVM-based model has also been developed to predict ionospheric scintillation at high latitudes, using a large variety of training features besides scintillation indices and spectral information, namely, geographical location, local time, solar wind and particle precipitation data, and geomagnetic indices. The persistence skill proved to be ineffective in predictions further than one hour [47]. Other machine learning algorithms, for example, Decision Tree and Random Forest, have also been applied in the detection of scintillation in ground-based measurements at the equator [48]. In the context of RO, machine learning has been previously applied to the detection of reflected signals [49,50].

The influence of the ionospheric scintillation in the F-layer on the bending angle standard deviation and the simulation of the effects observed in a limited set of three occultations when assuming a monotonic power law model has been previously investigated [42]. In this study, a data set of low-latitude occultations performed by the GNSS Receiver for Atmospheric Sounding (GRAS) on board of the Meteorological Operational (MetOp) satellites is created to train an automatic detection model of F-layer scintillations.

Given its previous application in the detection of ionospheric scintillation in GNSS signals [37,46], the performance of an SVM-based model to detect ionospheric scintillation in RO measurements is investigated in this study. Different training features are evaluated, such as amplitude and phase scintillation index and power spectral density, to define the classifier with the best overall performance in terms of accuracy, precision, recall and Receiver Operating Characteristic (ROC) curves. The main goal of the model is to automatically detect occultations containing typical spectral features related to ionospheric scintillation in the F-layer, i.e., with either monotonic or two-component power spectra in combination with the scintillation indices [27–36].

The remaining parts of this manuscript are structured as follows: the features assumed in the training of the classification model are detailed in Section 2. A description of the data set of low-latitude occultations curated for this study is addressed in Section 3. A brief description of the SVM algorithm as well as the different scenarios evaluated and performance metrics considered during the model training are discussed in Section 4. Finally, Sections 5 and 6 present the results and final remarks.

## 2. Ionospheric Characterisation

Given the geometry of occultation events, the GNSS signals are propagated through the ionosphere and the neutral atmosphere. Along the propagation path, the signal is refracted/bent as a response to the atmospheric refractivity. Assuming spherical symmetry, the accumulated bending angle along the ray path can be calculated using the phase component of the signal sampled in the LEO receiver, the occultation geometry and Snell's law [6,7]. Consequently, the refractivity at the ray tangent height can be retrieved from the bending angle profile [51].

The effects of the ionospheric refractivity, including regions of irregularities, are mostly observed within 30 km and the highest tangent altitude in RO measurements [52]. This range corresponds to the initial (final) and mostly flat segment of the setting (rising) occultation, where the ray tangents have not travelled through sections of the neutral atmosphere layer. Changes in the amplitude and phase of the GNSS signal in SLTA < 30 km are mostly caused by neutral atmosphere refractivity.

In order to automatically limit the segment of the occultation, a truncation threshold was set at 30 km straight-line tangent altitude (SLTA). The SNR segments were used to calculate features related to the signal amplitude, i.e., $S_4$ and intensity PSD. Similarly, disturbances in the excess phase were characterised by $\sigma_\phi$ and phase PSD. The amplitude and phase recorded by RO receivers, the same used in the retrieval of meteorological data, are available at UCAR/CDAAC's database in *atmPhs* files (level 1b).

### 2.1. Amplitude and Phase Indices

The amplitude scintillation is quantified by the $S_4$ index, which corresponds to the fourth moment of the received signal amplitude [2,53], normally computed as the standard deviation of the normalised signal intensity,

$$S_4 = \left[ \frac{\langle (I - \langle \bar{I} \rangle)^2 \rangle}{\langle \bar{I} \rangle^2} \right]^{1/2},$$  (1)

in which $I$ is the intensity of the RO signal, $\langle \, \rangle$ denotes the expectation operator and $\bar{I}$ is the filtered version of the signal intensity used as a reference in the calculation. $S_4$ index ranges between 0 and 1 with extreme cases over 1, indicating strong scatter and focusing phenomenon [33]. Different window lengths have been reported in the calculation of the averaged intensity, varying up to 60 s for ground-based measurements [37,46,48]. Given the shorter duration of RO measurements, a window length of 1 s (50 samples) was used in our calculation [54]. Filtering of the intensity reference is required to remove other sources of disturbances than ionospheric, i.e., the multipath effect, satellite motion, clock errors and thermal noise. A low-pass sixth-order Butterworth filter with a cut-off frequency of

0.1 Hz is commonly applied in this task [55]. The decision about the cut-off depends on the system orbit, and it can be critical in high-latitude measurements [56].

The effect of ionospheric irregularities in the phase is accounted for by $\sigma_\phi$, which is the standard deviation of the detrended excess phase [55]. The phase scintillation index is more frequently used in studies evaluating high-latitude measurements, given that the influence of the phase is more pronounced than in the amplitude in these regions [37,38]. The detrended excess phase was obtained using the same filtering approach applied to the $S_4$ calculation.

A common practice on the topic of scintillation detection is to rely on the values of amplitude and phase scintillation by defining threshold values as an initial indication that fluctuations in the signal were caused by ionospheric irregularities. Typical threshold values for $S_4$ and $\sigma_\phi$ are 0.2–0.3 and 0.1–0.2 rad, respectively [24,37,38,47]. Despite the low magnitude, the multipath is also quantified by the amplitude and phase scintillation indices [57]. Therefore, the indices alone can leave some margin of uncertainty about the source of the disturbance. Moreover, they provide insufficient information about the characteristics of the irregularities.

*2.2. Spectral Analysis*

The power spectral density is given by Fourier transform of the autocorrelation function of the signal intensity or phase [27,29]. Following the general formulation of the two-component model, the phase spectral density function (SDF) is defined as [32]

$$P(\mu) = \begin{cases} U_1|\mu|^{-p1}, & \mu \leq \mu_0 \\ U_2|\mu|^{-p2}, & \mu > \mu_0 \end{cases}, \tag{2}$$

where $p_{1,2}$ are the power spectral indices and $\mu_0$ is the normalised spectral break wavenumber. The wavenumber normalisation is given by $\mu_0 = q_0\rho_f$, where $q_0$ is the spectral break wavenumber and $\rho_f$ the Fresnel scale. The universal scattering strength can be represented as

$$U = \begin{cases} U_1, & \mu_0 \geq 1 \\ U_2, & \mu_0 < 1 \end{cases}, \tag{3}$$

with $U_2 = U_1\mu_0^{p_2-p_1}$. The formulation of the phase SDF assuming the scattering strength is convenient to define the two different scenarios of scatter, i.e., $U \ll 1$ weak and $U \gg 1$ strong scatter.

Assuming weak scatter and inverse monotonic power law with $U = U_1 = U_2$ and $p = p_1 = p_2$, then

$$U = C_s\rho_f^{(p-1)}, \tag{4}$$

and the phase PSD is reduced to

$$\Phi_{\delta\phi}(q) = C_s\, q^{-(p-1)}, \tag{5}$$

which defines the monotonic power law, where $C_s$ is the strength of the ionospheric turbulence, $p = 2\nu$, $\nu$ is the spectral slope and $q$ is the wavenumber [30]. Under weak scatter, the break scale wavenumber is often well defined in the intensity PSD. As a result of Fresnel filtering [39], the spectra present in most cases a flatness at low frequencies (long scale irregularities) and a roll-off starting at the maximum Fresnel frequency (break scale point),

$$f_b = \frac{V_s}{\sqrt{2\lambda d_s}},$$

(6)

in which $V_s$ is the effective scan velocity of the irregularities, $(2\lambda d_s)^{1/2}$ is the Fresnel radius, $\lambda$ is the radio signal wavelength and $d_s$ is the mean distance from the receiver to the region of irregularities according to the geometry of the system performing the measurements [34]. The PSD frequency scale ($f$) can be converted to wavelength scale, i.e., scale-length of the irregularities [35],

$$\lambda_s = \frac{V_s}{f}.$$

(7)

Given the geometry of the occultation events, determining the location of the region of irregularities along the ray path, i.e., $d_s$, is not a straightforward task [58]. Additionally, the scan velocity ($V_s$), related to the drift of the irregularities and to the orbit of the satellites, cannot be determined solely by occultations. Without complementary measurement of co-located systems, the intensity and phase power spectra can provide an estimation of the Fresnel frequency (under weak scatter) as well as an indication of strong scatter, following the occurrence of a two-component power law spectrum [32,40].

Figure 1 shows some examples of intensity and phase PSDs calculated from RO measurements. The PSDs were computed using Welch's method, assuming a Hamming window of 512 samples and 50% overlap, to obtain the averaged periodogram [36].

In Figure 1a,b, the disturbances in SNR shown in the left-most panels (blue curves) correspond to strong scintillation without saturation ($0.5 < S_4 < 1.0$). Disturbances are also observed in the excess phase (see detrended phase, central column). Vertical lines in the right-most panels show the estimated Fresnel maximum (spectral break), marking a clear transition between the flatness at the low-frequency end ($f < f_b$) and the asymptotic roll-off ($f > f_b$) in the intensity PSDs. Further, Figure 1b shows a phase PSD minimum around the possible Fresnel frequency ($f_b$), as described in [39]. Figure 1c shows similar characteristics even with most of the fluctuations observed in the first half of the signal amplitude and phase.

A case with saturated scintillation ($S_4 > 1$), likely related to focusing and strong scatter, is depicted in Figure 1d. The intensity PSD has an apparent two-component power law [32,35], in which the shallower slope ($p_1$) is related to the large-scale irregularities (outer scale, $\mu < \mu_0$), and the steeper slope ($p_2$) is related to small-scale irregularities (inner scale, $\mu > \mu_0$). The phase PSD does not follow the same trend (departure at high-frequency end) and most of the disturbance is observed after 15 s in the detrended phase. Figure 1e has fluctuations in SNR with a similar pattern to cases affected by a sporadic E-layer [59], with short duration and low intensity in this particular case. The intensity and phase PSD do not present the same characteristics as depicted in (a–d). The detrended phase is significantly different from the other cases, showing a large- rather than short-scale fluctuation pattern. Finally, Figure 1f corresponds to an occultation without signatures of disturbances, as observed in the SNR and detrended phase, and without the characteristics given in (a–d). Moreover, the occultations (e,f) are classified as low scintillation cases according to their scintillation indices ($S_4 \leq 0.2$).

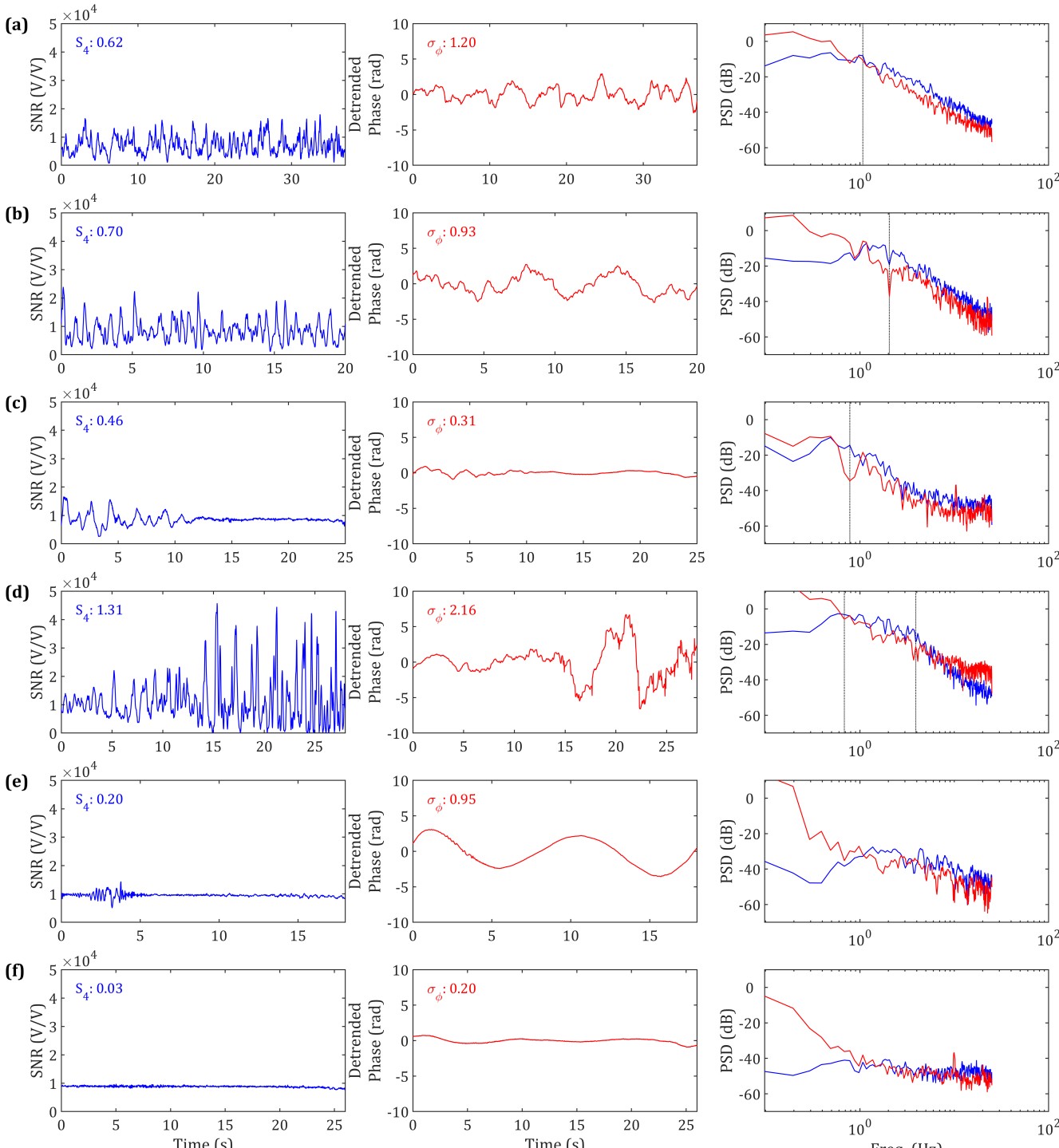

**Figure 1.** SNR, detrended excess phase, and intensity and phase PSDs. Blue and red curves are related to SNR, excess phase and their power spectra, respectively. MetOp measurements (**a**) MTPB.2015.079.03.41.G29, (**b**) MTPB.2015.079.01.58.G13 and (**c**) MTPB.2015.079.01.51.G19 show a monotonic power law in intensity and phase PSDs; (**d**) MTPB.2015.079.03.49.G23 depicts an apparent two-component power law in the intensity PSD, with phase PSD following a different trend at the high-frequency end; and (**e**) MTPA.2015.076.11.31.G07 and (**f**) MTPA.2015.077.03.17.G01 correspond to inconclusive cases or measurements not affected by disturbances in the SNR and detrended phase. Vertical lines show possible spectral breaks ($f_b$).

## 3. Data Set

### 3.1. Ionospheric Conditions

Ionospheric irregularities present distinct characteristics in different geographical regions, for example, low- and high-latitude, and they are also influenced by local time, season, solar and geomagnetic activity [4,40]. Measurements performed by ground-based receivers are limited in terms of geographic location but consist of hour-long recordings. GNSS-RO operates with global coverage, which offers the chance to record scintillations with more diversity regarding location by covering regions over sea and land. Despite the relatively short segment in which one occultation sounds mostly ionospheric effects, spectral analysis of RO measurements can be used to indicate the occurrence of ionospheric scintillation, as shown in Figure 1. Moreover, different GNSS-RO missions have been operating for more than 20 years. Therefore, the influence of different solar cycles and geomagnetic events and their influence in the ionosphere and in the RO data product can be investigated [17,21,22,24].

In order to develop a supervised detection model of F-layer scintillation in RO measurements, a data set of labelled observations is required. Given the different characteristics of scintillation regarding latitude, only measurements in the low-latitude region ($\pm 30°$ latitude) were selected in this study. High-latitude scintillations are known to affect the phase largely and, on occasion, exclusively, whereas amplitude is simultaneously affected at low latitudes [37] (and references therein). The occultations performed by MetOp-A and B satellites covered two different periods of the 24th solar cycle. The first interval consisted of occultations close to the solar cycle peak from 15 March (DOY 74) to 20 March (DOY 79) covering the St. Patrick's Day geomagnetic storm [22]. The second period corresponded to 14–19 May 2018 (DOY 134–139), around the minimum of the solar cycle. Besides the difference in the solar activity, the first period included days classified as disturbed regarding the level of geomagnetic activity owing to the St. Patrick's Day storm, whereas the second period corresponded to quiet days, according to Kp index. Figure 2 depicts the two intervals composing the data set during the 24th solar cycle and their respective levels of geomagnetic activity.

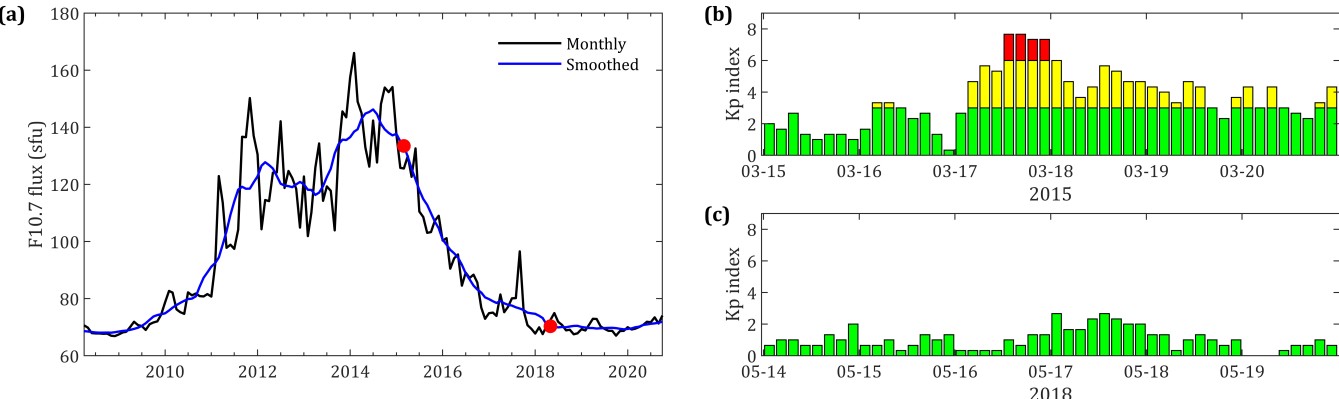

**Figure 2.** (**a**) Solar radio flux (F10.7) during the 24th solar cycle, which correlates to sunspot number. Red dots denote the periods considered in the data set. (**b,c**) Kp index with geomagnetic activity defined as low (green), medium (yellow) and high (red). The time series in 2015 shows medium and high Kp index values as a consequence of the St. Patrick's Storm. During 14–19 May 2018, days were classified as quiet (low activity) according to *Kp* index.

Complementary to the *Kp* index, the effects of the geomagnetic storm can also be observed through the values of the symmetric portion of the horizontal component magnetic field (SYM-H index), given in 1-minute resolution, and the interplanetary magnetic field (IMF) components, i.e., $B_x$, $B_y$ and $B_z$ (oriented along the poles). Figure 3 shows the indices recorded during the two periods considered in this study. In Figure 3a, the disturbances caused by the arrival of the solar winds during the early hours of 17 March turned the

SYM-H index negative and kept the same trend until around midnight on March 18th, reaching −226 nT as the minimum value. From this point, the SYM-H index indicated the start of the recovering phase, which continued until the last day in this interval. The disturbances were also registered by multiple flip points between positive and negative values in the IMF components. Figure 3g shows the IMF $B_z$ with a southward trend during most of March 17th, except for an abrupt northward turn between 06:00 UT and noon. The southward orientation indicates the coupling of the Earth's magnetosphere with the interplanetary magnetic field. In contrast to the effects of the St. Patrick Day's geomagnetic storm, Figure 3b,d,f,g show rather stable conditions during 14–19 May 2018, in which IMF components stayed within ±10 nT.

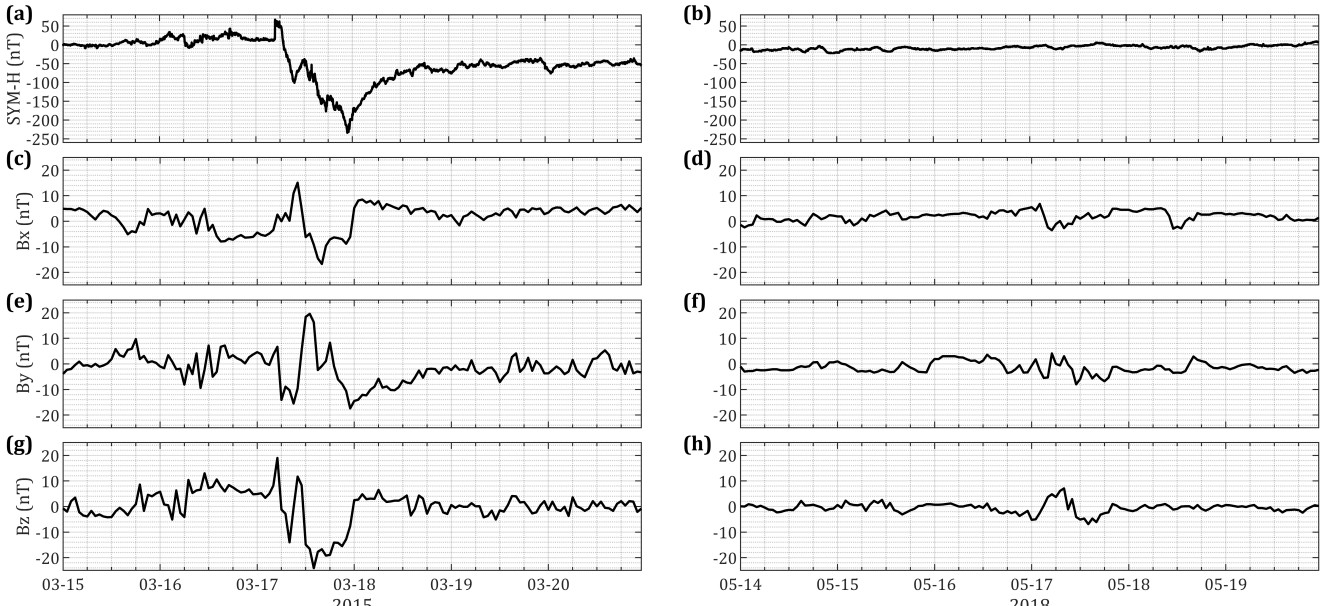

**Figure 3.** Geomagnetic records of (**a**,**b**) SYM-H indices, (**c**,**d**) IMF $B_x$, (**e**,**f**) IMF $B_y$ and (**g**,**h**) IMF $B_z$ during 14–20 March 2015 and 14–19 May 2018. Unit: nano-Tesla (nT).

### 3.2. Data Processing and Labelling

Some conditional checks were conducted to avoid measurements with anomalies and/or not of interest in the data set used in this study. As the first step, measurements without paired *atmPhs* and *atmPrf* files were not considered. The calculation of scintillation indices and power spectra requires amplitude and excess phase, given in *atmPhs* files. Moreover, the bending angle and refractivity profiles, given in *atmPrf*, are needed to investigate the effect of moderate and strong scintillation in RO data product in future studies. Next, occultations with signatures of scintillation caused by a sporadic E-layer, which creates U-shaped fades in the signal SNR with short duration, were not considered in this data set. E-layer scintillations were detected according to the procedure described in Zeng and Sokolovskiy [59].

Finally, a search for a maximum of three sections of the signal SNR with abrupt changes in the standard deviation was performed [60]. The standard deviation of the intervals, with a minimum length of 4 km, were computed, and occultations were removed whenever one of the sections had the standard deviation at least three times smaller than the interval with the greatest variation. This procedure eliminates occultations that were also potentially affected by sporadic E-layer scintillation but did not contain a deep fade in SNR and disturbances that are not present throughout the whole segment of the measurement [52]. The quasi-regular variations may correspond to a short interval in the occultation, insufficient to obtain statistical properties of ionospheric irregularities. The

steps detecting U-shaped fades and quasi-regular variations were overruled whenever $S_4 > 0.2$ corresponded to more than 50% of the occultation segments.

Features used in the characterisation of occultations were computed in the plateau region of each measurement, which were automatically truncated at 30 km SLTA. Below this tangent altitude, the neutral atmosphere contribution accounts for most of the effect observed in the signal [52]. The occultations with truncated segments shorter than 10 s (500 points) were also not considered in the study. Figure 4 shows some examples of measurements removed from the data set. Figure 4a represents one occultation with U-shaped fade related to sporadic E-layer scintillation, (b–e) correspond to different cases of quasi-regular disturbances and (f) represents a case with a plateau region shorter than 10 s.

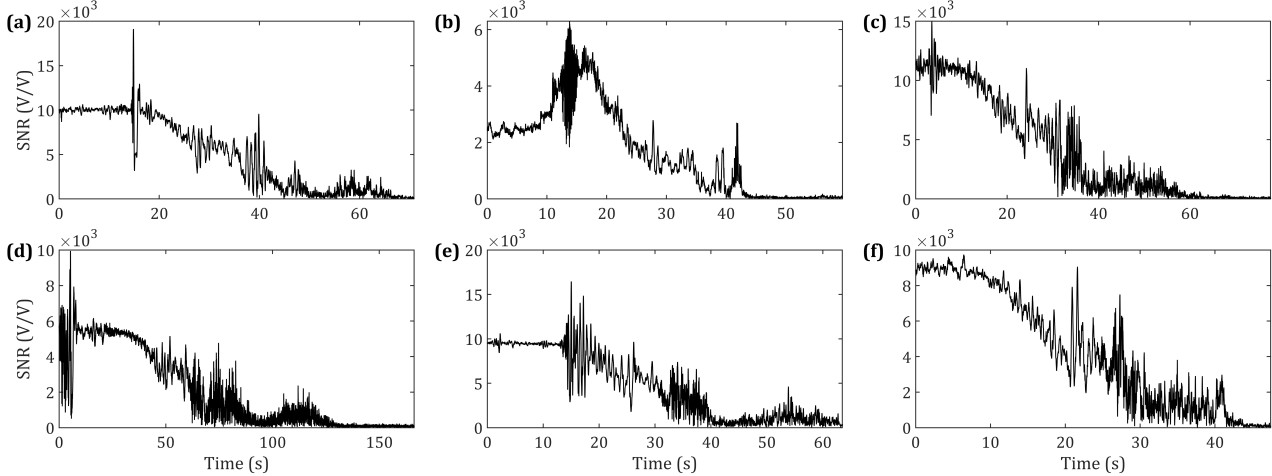

**Figure 4.** Removed RO measurements. (**a**) MTPA.2018.139.20.04.G01 shows a U-shaped scintillation signature; (**b**) MTPB.2015.076.05.01.G30, (**c**) MTPA.2018.134.20.08.G01, (**d**) MTPA.2018.134.21.50.G14 and (**e**) MTPA.2015.076.06.59.G12 depict cases with quasi-regular fluctuations (S-type) and (**f**) MTPB.2015.075.15.04.G31 depicts a case with plateau region shorter than 10 s.

Among 5340 low-latitude occultations, 436 measurements have been removed according to the conditional checks. The remaining 4904 occultations had their respective scintillation indices and power spectra computed based on their L1 C/A SNR and excess phase. Figure 5 shows the length of the occultation segment used in the analysis. The average of the segment length in MetOp-A/B measurement is 22.2 s, much shorter than the blocks evaluated in studies using measurements recorded by ground-based receivers [37,46].

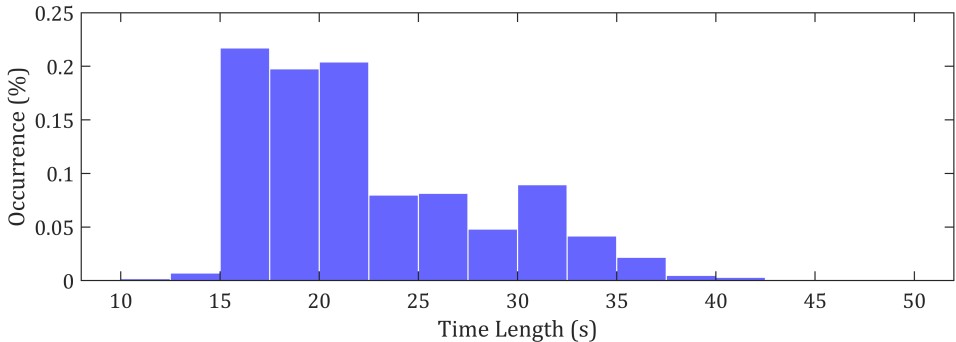

**Figure 5.** Length of the evaluated occultation segments. The average plateau length is around 22 s.

Amplitude scintillation was used as an initial step to sort out the data set as low ($S_4 \leq 0.2$), moderate ($0.2 < S_4 \leq 0.5$) and strong scintillation ($S_4 > 0.5$) [46]. The scintillation index provides an indication of disturbance related to ionosphere irregularities, which are more frequent during post-sunset hours and night-time [23,39]. Figures 6 and 7 show the

distribution of the different levels of scintillation observed in the data set in terms of local time (LT), according to the maximum $S_4$ index calculated in the occultation segment.

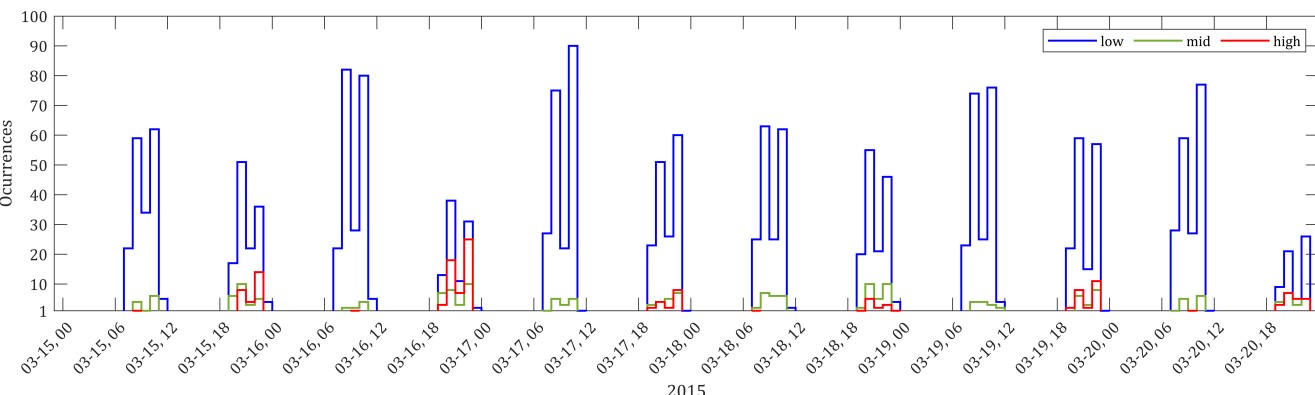

**Figure 6.** Local time distribution of $S_4$ index on MetOp-A,B occultations during the first period with high solar activity, i.e., 15 March (DOY 74) to 20 March (DOY 79) in 2015. "Low" corresponds to $S_4 \leq 0.2$, "Mid" to $0.2 < S_4 \leq 0.5$ and "High" to $S_4 > 0.5$. Strong scintillations were observed mostly during evenings.

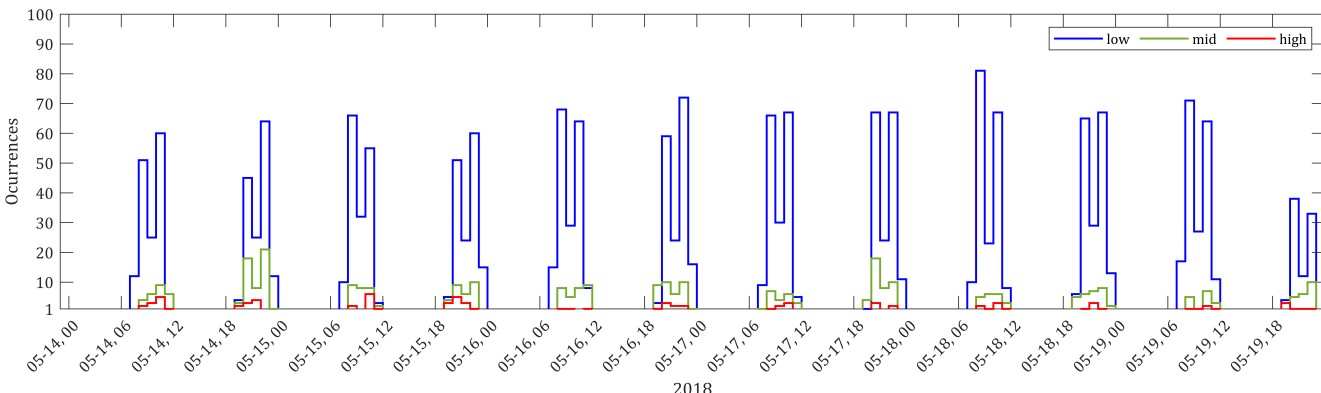

**Figure 7.** Local time distribution of $S_4$ index on MetOp-A,B occultations during the second period with low solar activity, i.e., 14 May (DOY 134) to 19 May (DOY 139) in 2018. "Low" corresponds to $S_4 \leq 0.2$, "Mid" to $0.2 < S_4 \leq 0.5$ and "High" to $S_4 > 0.5$. The occurrence of strong scintillations is lower than in the period evaluated in 2015.

The distribution of occultations shows their occurrence concentrated in two intervals according to local time. This is expected at the $\pm 30°$ latitude band given the sun-synchronous orbit of MetOp satellites, crossing the equator around 0930LT and 2130LT. Furthermore, the local minima at 0900LT and 2100LT bins are related to the absence of measurements at low latitudes [61].

In Figure 6, the computed scintillation indices covering the first period show a higher number of strong scintillations according to the $S_4$ index during 1800–0000 LT compared with 0600–1200 LT. Overall, less strong scintillations were found during the two days following the peak of the storm, i.e., March 18th and 19th. The same pattern was reported in the RO measurements performed by the FY3C mission [22] and by the ESA MONITOR Ionospheric Network, which described an increase of the TEC values during days 75 and 76 at low latitudes followed by a minimum on days 77 and 78 [62].

In Figure 7, strong scintillation occurrence was significantly reduced when compared to Figure 6, while more moderate cases were observed throughout the period. The reduction is likely related to the different level of solar activity registered during both periods composing this data set (see Figure 2) [4]. Common to both time-spans is the predominance of cases with low scintillation levels. About 83.03% and 82.01% of the occultations corresponded to $S_4 \leq 0.2$ during the periods evaluated in 2015 and 2018, respectively. Moderate

scintillations corresponded to 9.35% (2015) and 14.46% (2018) and strong disturbance levels were observed in 7.63% (2015) and 3.53% (2018) of the inspected occultations.

In this study, occultations with low scintillation have not been part of the data set used to train and test the automatic detection model. Low scintillation cases are mostly related to undisturbed and quiet ionospheric conditions and to the contributions of other sources of perturbation [46]. Moderate and strong scintillation cases had their intensity and phase power spectra visually inspected. Given exclusively low latitudes were considered, occultations were labelled as potential cases of ionospheric scintillation (label 1) whenever the intensity PSD showed a roll-off, which could be described by a single- or dual-slope power law. Phase PSDs were used as complementary information, confirming the indication of a typical disturbance caused by ionospheric irregularities when its slope followed closely the intensity PSD. Cases not reaching those criteria were labelled as occultations in which the disturbance was potentially not caused by ionospheric irregularities or were inconclusive (label 0, other disturbances). Accordingly, the occultations shown in Figure 1a–d correspond to label 1 cases.

The detection of cases labelled as "1" are relevant for the task of characterising the regions of irregularities and to eventually replicate the effects in GNSS signals in simulations by estimating the parameters defining the power law observed in the power spectral analysis. The relation between such parameters and the influence on the RO product is also a potential aspect to be further investigated.

Figures 8 and 9 show the distribution of labelled cases according to local time during the periods in 2015 and 2018. The two intervals correspond to 858 measurements, with 413 label 1 cases and 445 label 0. This corresponds to a ratio of 1:1.08 between classes and, therefore, the data set is balanced. A comparison indicates that label 1 cases were more frequent during the period covering different phases of the St. Patrick's Day storm, also corresponding to an interval around the maximum of the solar cycle. Apart from DOY 77 and 78, a significantly higher amount of label 1 cases was found than label 0. Regarding the interval in 2018, there was a majority of label 0 cases throughout the days evaluated. Label 0 corresponded to 38.52% (2015) and 63.80% (2018), whereas label 1 corresponded to 61.48% (2015) and 36.20% (2018) of the observations.

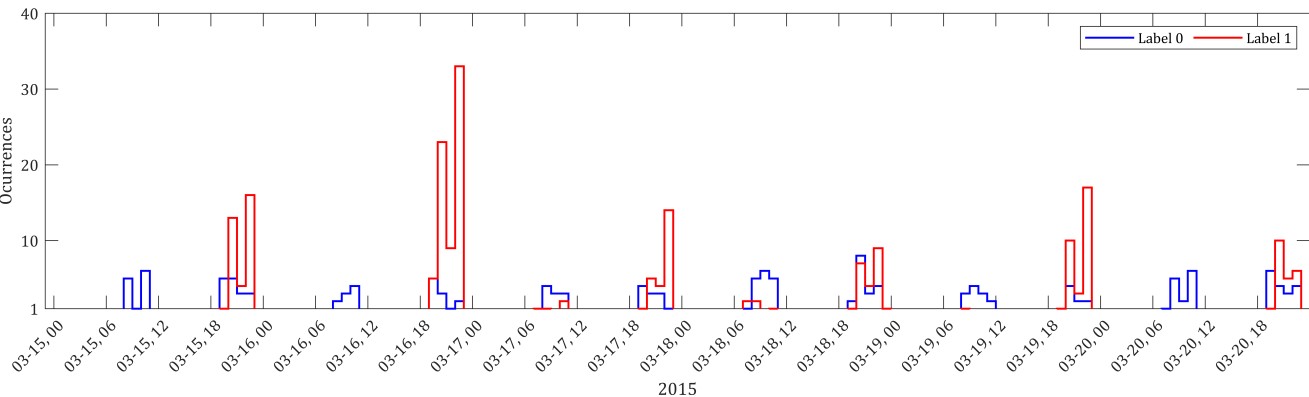

**Figure 8.** Local time distribution of labelled occultations between March 15th (DOY 74) and March 20th (DOY 79) in 2015. Label 1 observations correspond to the majority of the observations composing the data set in this period, likely related to the solar cycle period.

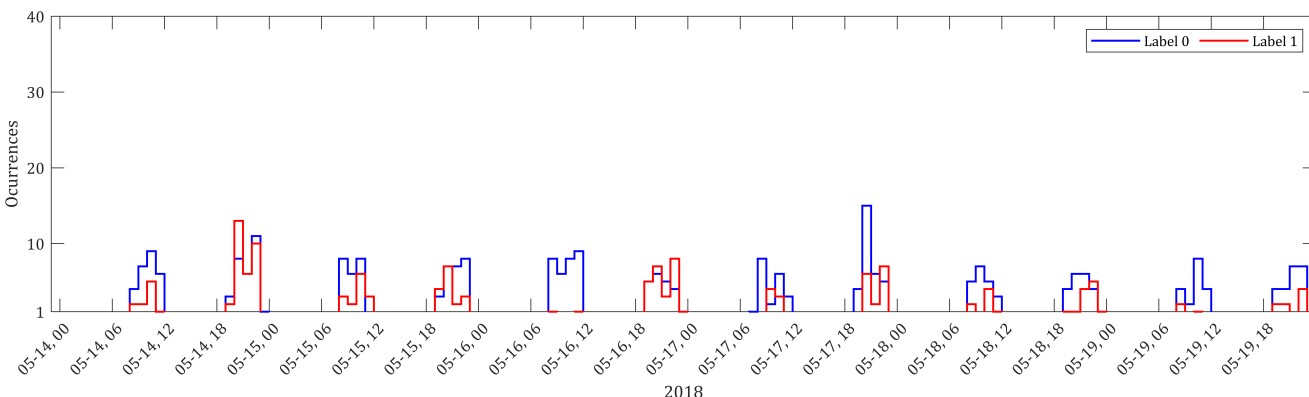

**Figure 9.** Local time distribution of labelled occultations between 14 May (DOY 134) and 19 May (DOY 139) in 2018. Label 0 observations are the majority of the observations in this period, alternating the trend observed in March 2015.

Figure 10 shows the daily averaged PSDs for label 1, with blue curves corresponding to the intensity PSD and red to the phase spectra. The power spectral indices for different days were estimated with least squares fit in the logarithmic domain. The frequency interval was set manually for each day with the lower bound being the spectral break limiting the flat part ($f < f_b$) and the upper bound being either the highest frequency of the spectra or the noise floor (flatness in high-frequency end). The averaged spectral indices were within 2.77 and 3.37 with the phase spectra followed closely by the intensity asymptote. Observations with monotonic power laws were the majority in comparison to the number of occultations with an apparent two-component power law, which explains only single slopes being observed in the averaged PSDs. The power spectral indices agreed with the values expected for scenarios of scintillation, with values tending to three as expected for the F-layer scintillation in low-latitude [27,33,39,55].

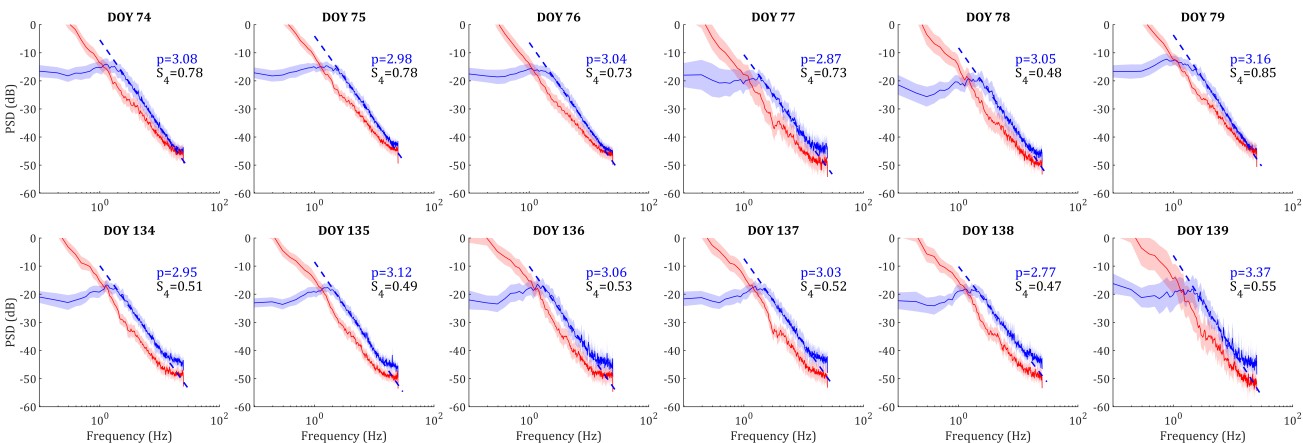

**Figure 10.** Daily averaged PSDs for label 1 observations. Blue curves correspond to intensity, red corresponds to the phase, and shaded regions to the 95% confidence interval. Spectral slopes estimated by least squares fit within 2.77 and 3.37, with phase slopes closely following the trend. The $S_4$ index corresponds to the average among label 1 cases in each day.

Figure 11 shows the daily averages for label 0 observations, with shallower slopes compared to label 1 occultations. This aspect was expected since the absence of a clear asymptote, for example, disturbance potentially not caused by the ionospheric irregularities, or coarse power spectra would result in shallower spectral indices compared to label 1 daily averages. Moreover, the phase spectra showed a similar pattern among different days, i.e., a shallow curve at the high-frequency end. Such a pattern escapes from the common asymptote observed in intensity and phase in Figure 10. In addition, Label 0 cases are expected to be most frequently associated with lower $S_4$ indices than label 1 cases. The

presence of an outer scale given $p < 3$ should not lead to saturation ($S_4 > 1$) [30–32]. The daily average $S_4$ indices show overall higher values in label 1 than in label 0 occultations.

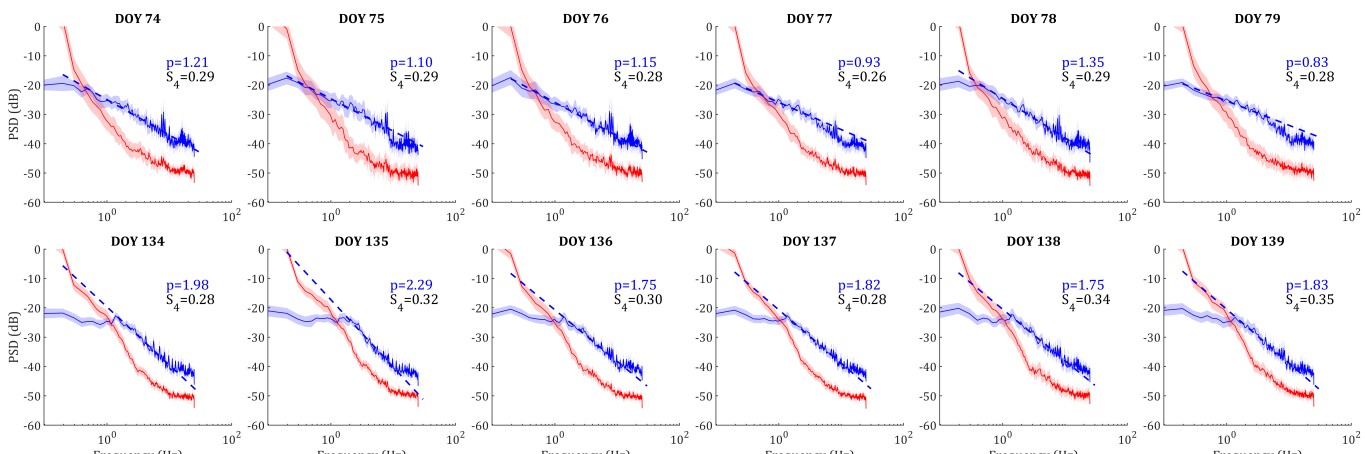

**Figure 11.** Daily averaged PSDs for label 0 observations. Blue curves correspond to intensity, red corresponds to phase, and shaded regions to the 95% confidence interval. Spectral slopes are in general shallower than the ones observed in the averaged intensity PSDs of label 1 measurements. Phase PSDs do not follow closely the intensity slopes at the high–frequency end. The $S_4$ index corresponds to the average among label 0 cases in each day.

Figure 12 shows the distribution of maximum $S_4$ and $\sigma_{\phi\text{avg}}$ for label 0 and label 1 combining occultations in 2015 and 2018. Label 0 is predominant regarding the $S_4$ index within $0.2 < S_4 < 0.4$ and $0 < \sigma_{\phi\text{avg}} < 0.4$ rad. The left-most column presents the bending angle standard deviation (STDV) of observations performed between 0930–2130 LT and 2130–0930 LT. This parameter is computed within a 60–80 km impact height and, among different sources of disturbance, quantifies the small-scale ionospheric irregularities [63]. The *STDV* mean values in label 1 observations were higher than in label 0.

An overview of the criteria used in the preparation of the data set is presented in Table 1, including the withdrawal conditions and the two classes considered in this study.

**Table 1.** Conditional checks and classes assumed during data set labelling.

| Criteria | Description | Label |
|---|---|---|
| $S_{4max} \leq 0.2$ | Low-scintillation cases | (Removed) |
| Sporadic E-layer [59] | Occultations with U-shaped fade and $S_4 \leq 0.2$ corresponding to less than 50% of the plateau | (Removed) |
| S-type disturbances [52] | Occultation with quasi-regular disturbances and $S_4 \leq 0.2$ corresponding to less than 50% of the plateau | (Removed) |
| Others | PSD without clear trend of monotonic or double-slope inverse power law or inconclusive PSD | 0 |
| | PSD with trend of monotonic or double-slope inverse power law | 1 |

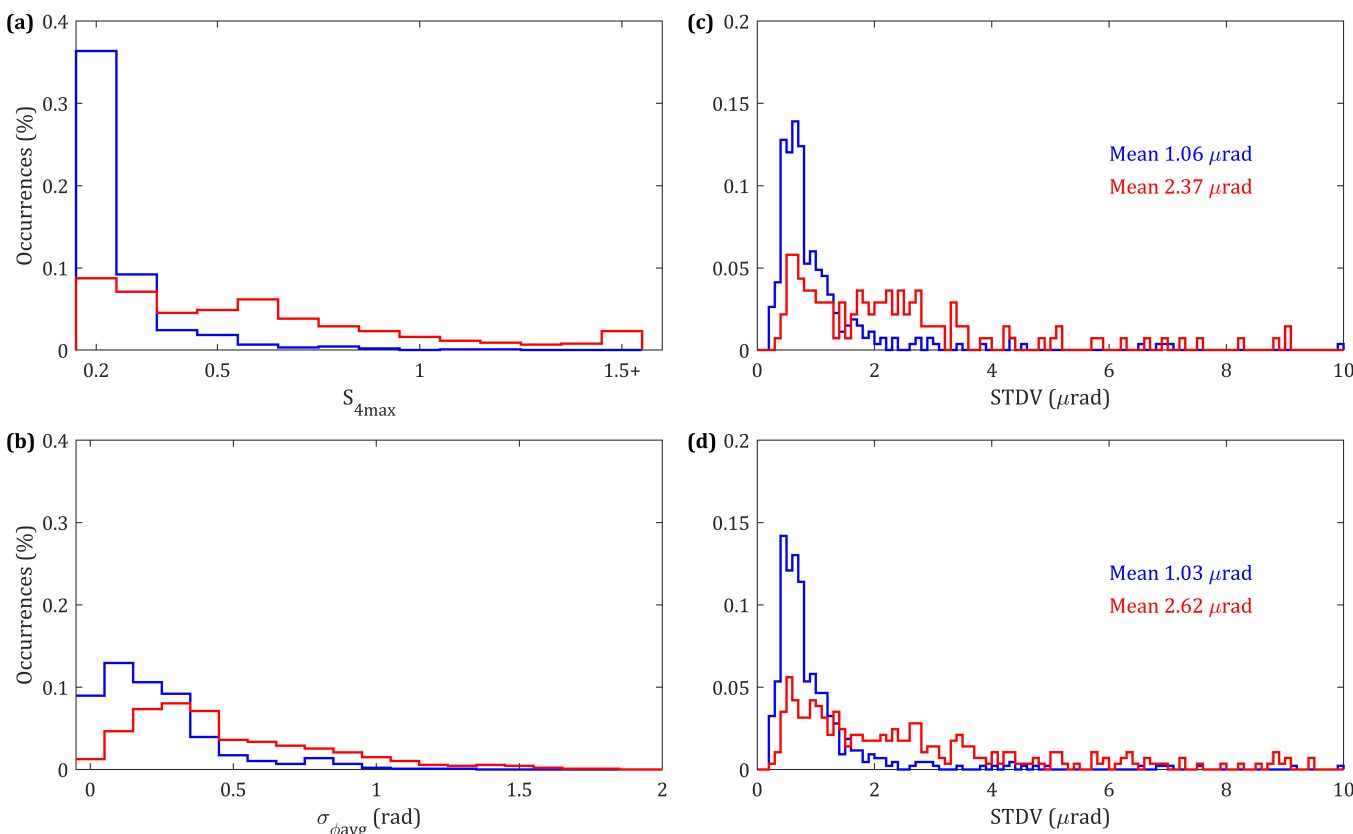

**Figure 12.** Distribution of (**a**) maximum $S_4$ index and (**b**) $\sigma_{\phi\,\text{avg}}$ (2015 and 2018 accumulated): Label 1 occultations are predominant in moderate and high scintillation indices; STDV distribution between (**c**) 0930–2130 LT and (**d**) 2130–0930 LT: Mean STDV values are significantly higher in label 1 observations than in label 0 in both day periods. Label 0 and 1 correspond to blue and red lines, respectively. Occultations with $STDV > 10\,\mu$rad were not considered in the figure.

## 4. Support Vector Machine

The data set of occultations was used in the training of the Support Vector Machine (SVM) algorithm [44] to perform automatic classification between the two predefined classes: occultation with characteristics of ionospheric scintillation (label 1) and other disturbances (label 0). The same algorithm has been applied to create detection models of low- and high-latitude ionospheric scintillation trained on data sets composed by ground-based measurements [37,46,47]. In short, the supervised algorithm attempts to find common characteristics between the training observations (labelled) and new occultations (unlabelled).

The SVM algorithm finds the hyperplane function, which separates the occultation between the two classes by minimising the cost function

$$\min_\theta C \sum_{i=1}^{m} \left[ y^{(i)} \,\text{cost}_1\left(\theta^T k^{(i)}\right) + \left(1 - y^{(i)}\right) \text{cost}_0\left(\theta^T k^{(i)}\right) \right] + \frac{1}{2} \sum_{j=1}^{n} \theta_j^2, \tag{8}$$

where $k^{(i)}$ is the value of the kernel function for the $i$-th vector of training features, $y^{(i)}$ is the labelled class corresponding to the $x^i$, $\theta_j$ are the $j$-th coefficient of the hyperplane function, and $C$ is the penalty or box constraint parameter. Functions $\text{cost}_{1,0}$ are defined as

$$\text{cost}_1 = -\log\left(h_\theta\left(k^{(i)}\right)\right), \tag{9}$$

$$\text{cost}_0 = -\log\left(1 - h_\theta\left(k^{(i)}\right)\right), \tag{10}$$

with $h(x) = 1/(1 + \exp(-\theta^T x))$, the Sigmoid function. The optimised coefficients define the hyperplane function

$$g(x) = \theta^T x + b \quad \begin{cases} \text{label 1,} & \text{if } g(x) \geq 1 \\ \text{label 0,} & \text{if } g(x) \leq -1 \end{cases} \text{,} \tag{11}$$

with maximum margin to the closest training points. The tolerance of this margin is controlled by the hyperparameter $C$. A thorough definition of the method is given in [46].

The standard definition of the SVM algorithm is given, assuming that the classes are separable by a linear function (kernel). Thus,

$$\mathbf{k} = [x^{(1)} \ x^{(2)} \ \dots \ x^{(n)}], \tag{12}$$

where each element in the vector corresponds to one feature. Alternatively to the linear kernel form, SVM has as an advantage the possibility of using different kernel functions, especially in problems with high complexity [45]. The kernel function is equivalent to mapping the problem to a higher dimensional feature space while still defining the hyperplane function in the original low-dimensional space [46]. A common function, also evaluated in this study, is the radial basis function (RBF) or Gaussian kernel,

$$k^{(i)} = \exp\left(-\frac{||x - x^{(i)}||^2}{2\gamma^2}\right), \tag{13}$$

in which $\gamma$ is equivalent to the width of the Gaussian kernel, i.e., the standard deviation [64].

### 4.1. Feature Selection

Among the parameters extracted from MetOp-A/B measurements, the performance of the amplitude and phase features was evaluated separately to have a clear indication of the contribution of the gradual addition of features to the training vector. The scenarios involving amplitude features were:

- Maximum $S_4$ during the occultation segment (1 feature);
- Maximum and mean $S_4$ during the occultation segment (2 features);
- Intensity PSD (257 features);
- Maximum and mean $S_4$, and intensity PSD (259 features).

Despite the most common application for quantifying scintillation in high-latitude measurements [37], scenarios involving the following (excess) phase features were also investigated:

- Maximum $\sigma_\phi$ during the occultation segment (1 features);
- Maximum and mean $\sigma_\phi$ during the occultation segment (2 features);
- Phase PSD (257 features);
- Maximum and mean $\sigma_\phi$, and phase PSD (259 features).

Additionally, scenarios combining the amplitude and phase features were also evaluated. The step of feature selection was performed using linear and Gaussian kernels.

### 4.2. Performance Evaluation

In order to evaluate the classification performance of the trained models (hypothesis), a portion of the data set not used during the training step was fed into the classifier. The accuracy achieved during tests (cross-validation) is defined as

$$\text{Accuracy} = \frac{\text{TN} + \text{TP}}{\text{TN} + \text{FP} + \text{FN} + \text{TP}}, \tag{14}$$

where TN is the number of occultations correctly labelled as "0", FP is the number of false positives, FN corresponds to the number of false negatives, and TP stands for the number

of occultations correctly classified as "1". Figure 13 shows the confusion matrix, which is composed of the described terms.

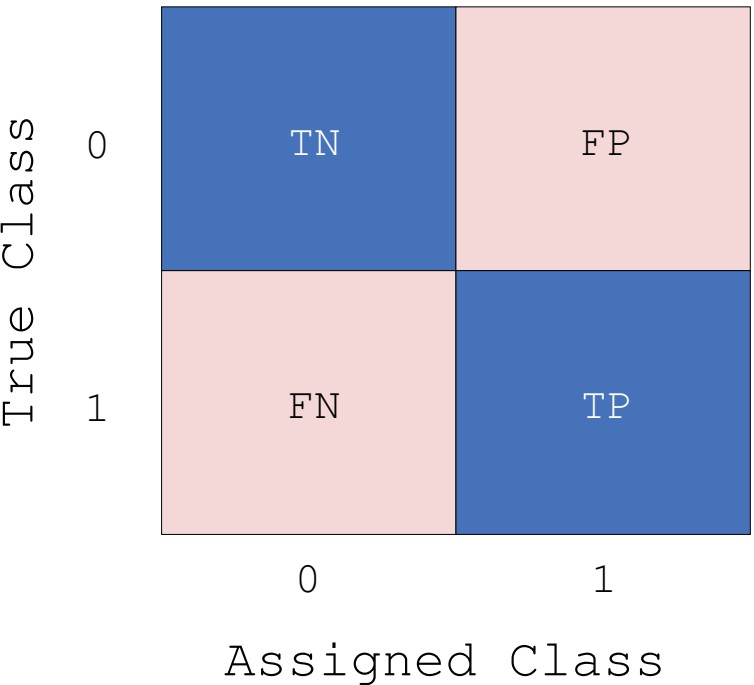

**Figure 13.** Confusion (contingency) matrix.

An F-score has been used as a complementary metric to accuracy, and it can be described as a harmonic mean of precision and recall,

$$\text{F-score} = \frac{2(\text{Precision} \cdot \text{Recall})}{\text{Precision} + \text{Recall}}. \tag{15}$$

Precision quantifies the accuracy of the classifier in marking potential cases of scintillation as label 1, and it is given by the ratio between the number of correctly assigned labels and the total number of occultations in such class,

$$\text{Precision} = \frac{\text{TP}}{\text{TP} + \text{FP}}. \tag{16}$$

Recall indicates the ability of the classifier to distinguish the two classes. In the context of this study, it indicates the number of occultations potentially affected by ionospheric scintillation misclassified as label 0,

$$\text{Recall} = \frac{\text{TP}}{\text{TP} + \text{FN}}. \tag{17}$$

Ideally, the classifier should reach high precision and recall. Thus, the F-score should capture the poor performance of the trained classifier either in one or in both tasks. Furthermore, Receiver Operating Characteristics (ROC) curve has also been considered to compare the true positive rate (TPR) or recall and false positive rate (FPR) among the different scenarios evaluated during the selection of the classifier [65]. FPR, also known as fall-out, is defined as

$$\text{FPR} = \frac{\text{FP}}{\text{N}} = \frac{\text{FP}}{\text{FP} + \text{TN}}, \tag{18}$$

where N is the total number of label 0 observations.

Given the relative small number of occultations composing the data set, *k*-fold cross-validation was performed to train and evaluate the classification performance of each of

the scenarios under investigation. In the cross-validation, the full data set was split into 10 stratified folds ($k = 10$), i.e., with an equal ratio of classes in each fold (stratified). Thus, nine infolds were used to train the SVM algorithm and one outfold was used to test the performance of the model. This procedure was repeated 10 times, with a permutation of the outfold in every training iteration. Figure 14 shows an overview of the steps carried out during the evaluation of the supervised detection model including occultation preprocessing, labelling, and training and testing of the SVM algorithm.

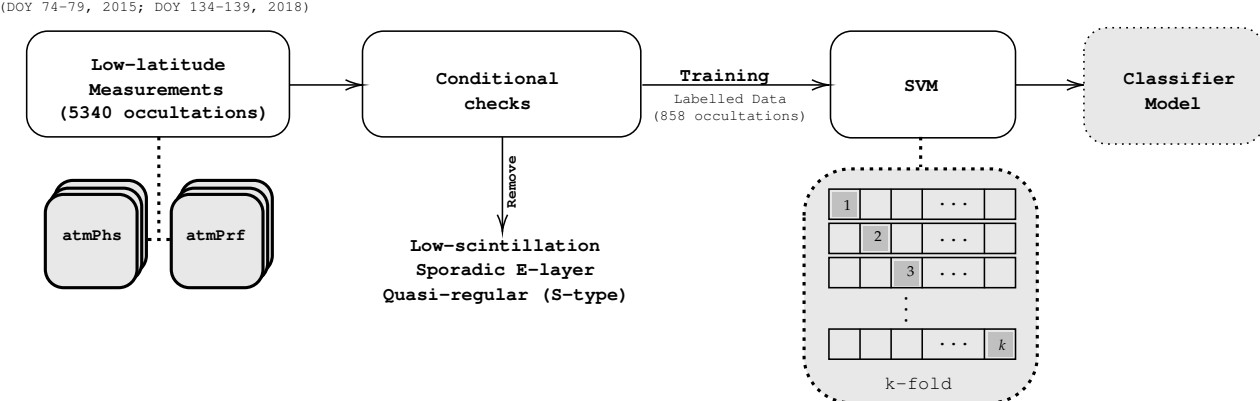

**Figure 14.** Overview of preprocessing, labelling, and training and testing steps.

## 5. Results

The hyperparameters $C$ and $\gamma$ were optimised with Bayesian optimisation. The search spaces (in log-scale) were limited to $[1 \times 10^{-5}, 1 \times 10^{5}]$ for both hyperparameters. The same seed was used to select the different observations composing the folds for all scenarios evaluated. The metrics and the ROC curves are presented as the averages of the test folds.

*Feature Selection*

Figure 15 show the ROC curves obtained with the different sets of training features, with amplitude and phase features investigated separately. The dots correspond to the operating points, which were defined by assuming the same penalty between classes.

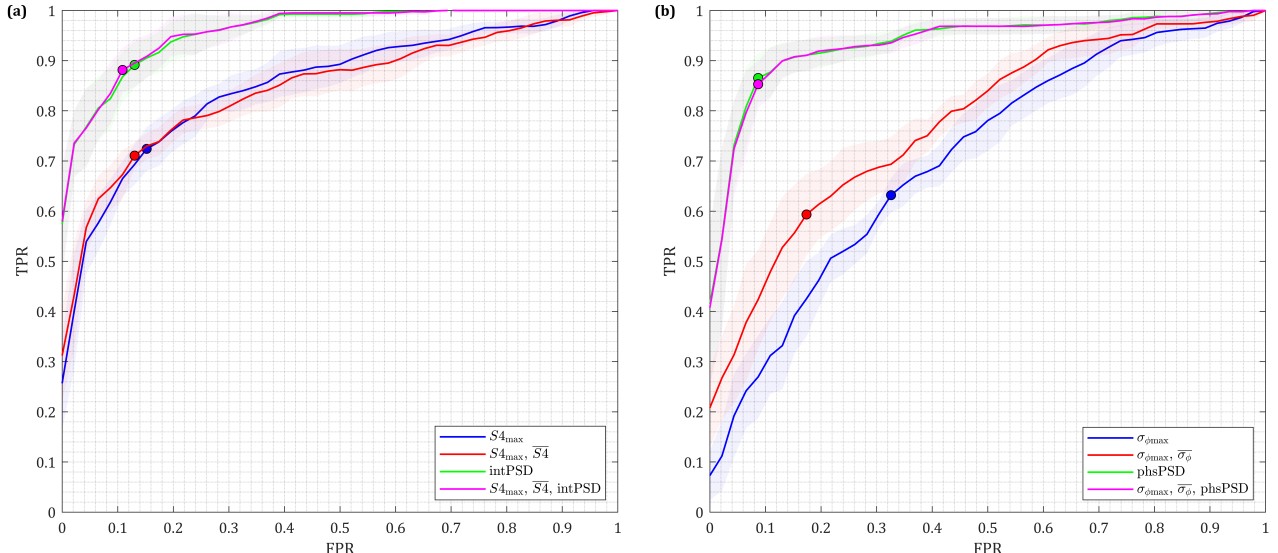

**Figure 15.** ROC curves obtained with a linear kernel for scenarios assuming (**a**) amplitude and (**b**) phase features. Shaded regions correspond to the 95% confidence interval.

Amplitude and phase scintillation indices showed the worst performance when assumed as the only training features. Amplitude scintillation index outperformed $\sigma_\phi$ in all metrics. The models trained only with the phase index have performed slightly better than random classification. In Figure 15a, the intensity PSD contributed to the significant increase in the level of true positives. In Figure 15b, the classifiers trained with phase PSD features achieved similar levels of TPR to the best amplitude scenarios with smaller FPR, besides less variation between folds. In general, $S_4$ and $\sigma_\phi$ had marginal contributions to the PSDs in the overall performance. Table 2 summarises the performance of the models obtained with linear kernel.

**Table 2.** Performance overview with optimised linear kernel.

| Training Vector | Accuracy | Precision | Recall | F-Score | TPR | FPR |
|---|---|---|---|---|---|---|
| $S_{4max}$ | $0.767 \pm 0.017$ | $0.849 \pm 0.029$ | $0.629 \pm 0.037$ | $0.721 \pm 0.025$ | 0.724 | 0.152 |
| $S_{4max}, \overline{S_4}$ | $0.770 \pm 0.019$ | $0.901 \pm 0.023$ | $0.588 \pm 0.044$ | $0.710 \pm 0.033$ | 0.711 | 0.130 |
| intPSD | $0.878 \pm 0.026$ | $0.897 \pm 0.026$ | $0.842 \pm 0.051$ | $0.868 \pm 0.032$ | 0.892 | 0.130 |
| $S_{4max}, \overline{S_4}$, intPSD | $0.881 \pm 0.025$ | $0.899 \pm 0.026$ | $0.850 \pm 0.048$ | $0.872 \pm 0.030$ | 0.881 | 0.109 |
| $\sigma_{\phi max}$ | $0.623 \pm 0.023$ | $0.710 \pm 0.065$ | $0.385 \pm 0.049$ | $0.494 \pm 0.039$ | 0.623 | 0.326 |
| $\sigma_{\phi max}, \overline{\sigma_\phi}$ | $0.712 \pm 0.038$ | $0.766 \pm 0.060$ | $0.584 \pm 0.053$ | $0.660 \pm 0.047$ | 0.593 | 0.174 |
| phsPSD | $0.885 \pm 0.019$ | $0.890 \pm 0.025$ | $0.870 \pm 0.030$ | $0.879 \pm 0.020$ | 0.865 | 0.087 |
| $\sigma_{\phi max}, \overline{\sigma_\phi}$, phs PSD | $0.885 \pm 0.019$ | $0.886 \pm 0.028$ | $0.874 \pm 0.026$ | $0.879 \pm 0.020$ | 0.853 | 0.087 |

Figure 16 shows the ROC curves for models trained with Gaussian kernel and the same set of training vectors evaluated with the linear kernel.

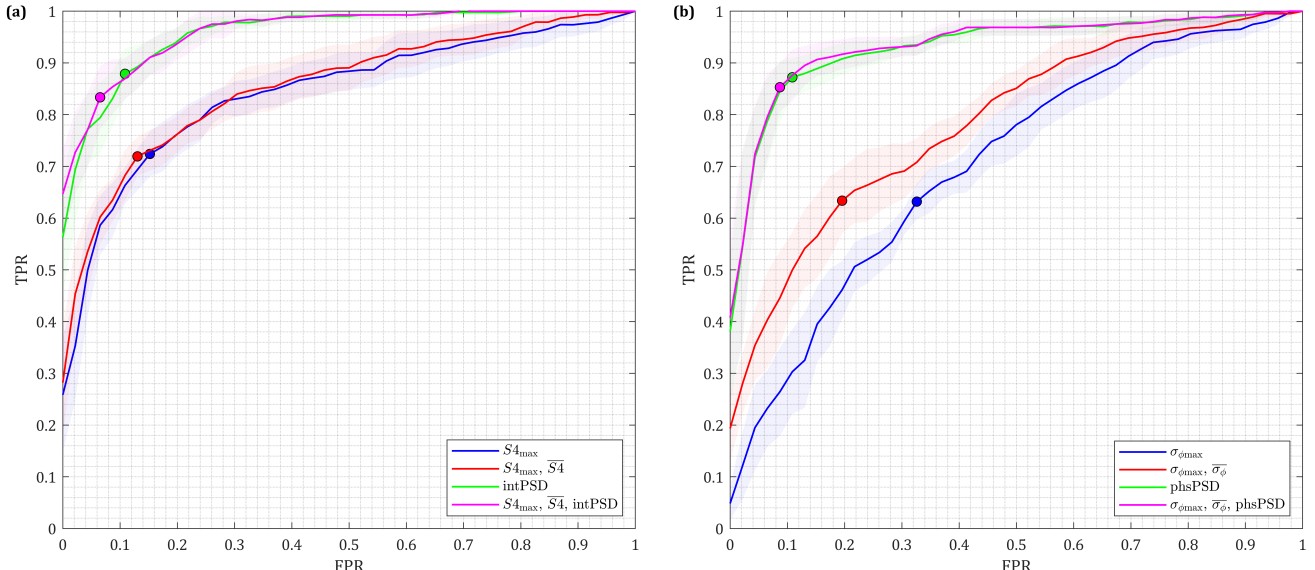

**Figure 16.** ROC curves obtained with Gaussian kernel for scenarios assuming (**a**) amplitude and (**b**) phase features. Shaded regions correspond to the 95% confidence interval.

The most noticeable contribution of the Gaussian kernel is observed in terms of recall and, consequently, F-score when only scintillation indices were used as training features, especially $\sigma_\phi$. However, there was no improvement in the scenarios considering PSD. Table 3 summarises the performance with the Gaussian kernel.

**Table 3.** Performance overview with optimised Gaussian kernel.

| Training Vector | Accuracy | Precision | Recall | F-Score | TPR | FPR |
|---|---|---|---|---|---|---|
| $S_{4max}$ | $0.782 \pm 0.019$ | $0.837 \pm 0.035$ | $0.683 \pm 0.029$ | $0.751 \pm 0.021$ | 0.724 | 0.152 |
| $S_{4max}, \overline{S_4}$ | $0.783 \pm 0.013$ | $0.852 \pm 0.026$ | $0.688 \pm 0.032$ | $0.747 \pm 0.019$ | 0.719 | 0.130 |
| intPSD | $0.883 \pm 0.022$ | $0.888 \pm 0.038$ | $0.872 \pm 0.038$ | $0.878 \pm 0.024$ | 0.879 | 0.109 |
| $S_{4max}, \overline{S_4}$, intPSD | $0.881 \pm 0.020$ | $0.891 \pm 0.024$ | $0.860 \pm 0.042$ | $0.874 \pm 0.024$ | 0.833 | 0.065 |
| $\sigma_{\phi max}$ | $0.646 \pm 0.023$ | $0.645 \pm 0.039$ | $0.605 \pm 0.035$ | $0.682 \pm 0.018$ | 0.632 | 0.326 |
| $\sigma_{\phi max}, \overline{\sigma_\phi}$ | $0.716 \pm 0.039$ | $0.718 \pm 0.049$ | $0.680 \pm 0.047$ | $0.697 \pm 0.040$ | 0.634 | 0.196 |
| phsPSD | $0.883 \pm 0.020$ | $0.887 \pm 0.028$ | $0.869 \pm 0.025$ | $0.878 \pm 0.021$ | 0.872 | 0.109 |
| $\sigma_{\phi max}, \overline{\sigma_\phi}$, phs PSD | $0.883 \pm 0.018$ | $0.884 \pm 0.029$ | $0.874 \pm 0.026$ | $0.878 \pm 0.019$ | 0.853 | 0.087 |

Lastly, the combination of amplitude and phase training features were also evaluated with linear and Gaussian kernels. The assessment of intensity and PSD was compared with the addition of $S_4$ maximum and mean and, finally, $\sigma_\phi$ maximum and mean. Figure 17 shows the ROC curves for these scenarios.

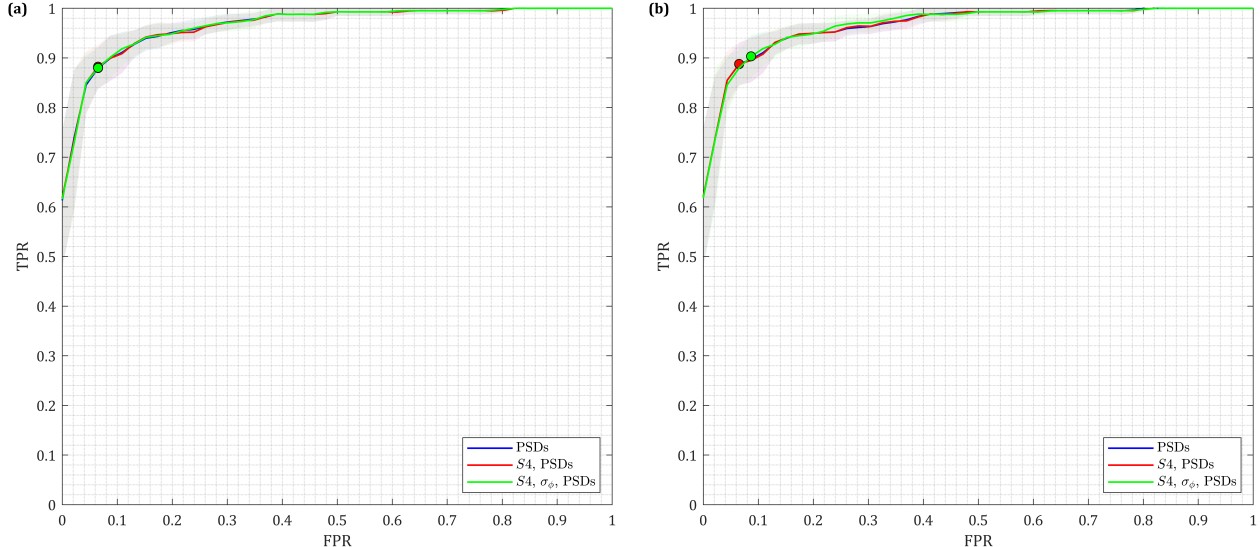

**Figure 17.** ROC curves for the combination of amplitude and phase training features obtained with (**a**) linear and (**b**) Gaussian kernel. Shaded regions correspond to the 95% confidence interval.

The combination of intensity and phase power spectra improved accuracy, precision and recall, while it reduced the number of observations wrongly classified (lower FPR). The addition of amplitude and phase scintillation indices had no contribution to performance. Regarding the kernel functions, the results are quite similar. Therefore, the model trained with the combination of amplitude PSDs and linear kernel is the most reasonable choice. The threshold value at the operating point was 0.595. Table 4 summarises the results of combining the amplitude and phase features and the respective optimised hyperparameters.

Figure 18 shows the confusion matrices for the test folds trained with amplitude and phase PSDs and assuming a linear kernel. The matrices were used in the calculation of the performance metrics.

**Table 4.** Performance overview combining amplitude and phase features.

| Training Vector | $C$ | $\gamma$ | Accuracy | Precision | Recall | F-Score | TPR | FPR |
|---|---|---|---|---|---|---|---|---|
| PSDs | 0.005 | - | $0.910 \pm 0.019$ | $0.920 \pm 0.026$ | $0.895 \pm 0.040$ | $0.905 \pm 0.021$ | 0.880 | 0.065 |
| $S_4$, PSDs | 0.006 | - | $0.913 \pm 0.018$ | $0.922 \pm 0.021$ | $0.896 \pm 0.039$ | $0.906 \pm 0.022$ | 0.880 | 0.065 |
| $S_4$, $\sigma_\phi$, PSDs | 0.006 | - | $0.910 \pm 0.019$ | $0.920 \pm 0.026$ | $0.894 \pm 0.040$ | $0.905 \pm 0.021$ | 0.888 | 0.065 |
| PSDs | 72 693 | 4 314 | $0.910 \pm 0.021$ | $0.915 \pm 0.026$ | $0.896 \pm 0.038$ | $0.904 \pm 0.024$ | 0.882 | 0.065 |
| $S_4$, PSDs | 2 597 | 1 019 | $0.910 \pm 0.020$ | $0.916 \pm 0.030$ | $0.898 \pm 0.039$ | $0.906 \pm 0.022$ | 0.887 | 0.065 |
| $S_4$, $\sigma_\phi$, PSDs | 47 637 | 3 560 | $0.909 \pm 0.020$ | $0.917 \pm 0.026$ | $0.894 \pm 0.038$ | $0.904 \pm 0.023$ | 0.903 | 0.087 |

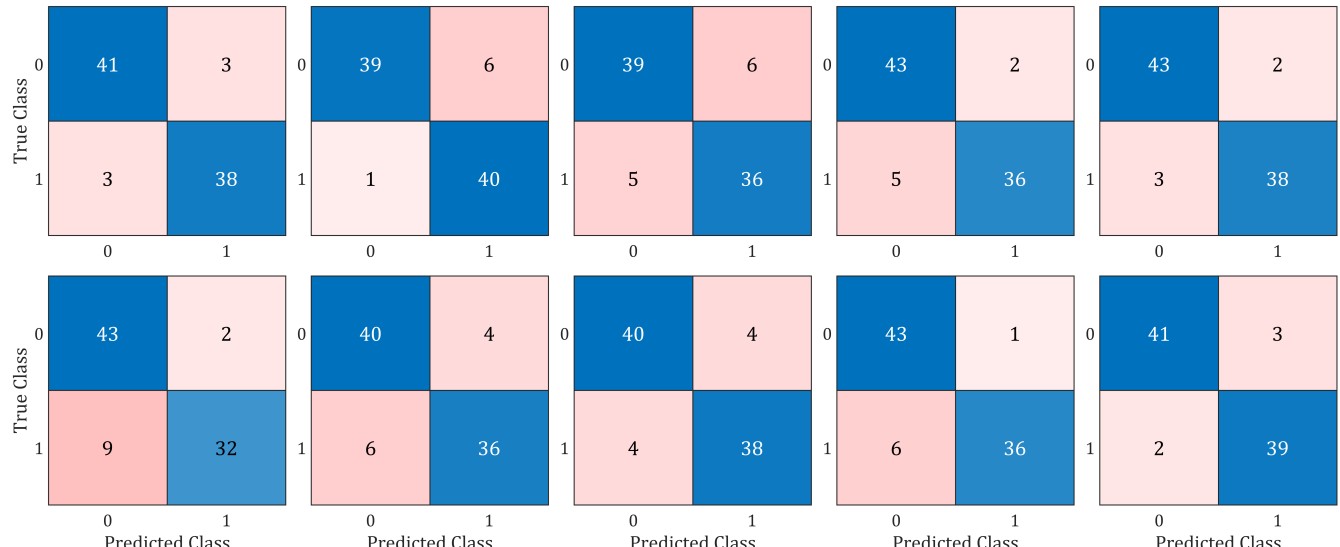

**Figure 18.** Confusion matrices for test folds: intensity and phase PSDs as training features and linear kernel.

## 6. Conclusions

In this study, GNSS-RO measurements performed by MetOp-A/B at low latitudes during two intervals of the 24th solar cycle have been analysed. The first interval corresponded to the days before, during and after the St. Patrick's Day geomagnetic storm in 2015. The second interval, during May 2018, corresponded to days with low geomagnetic activity in addition to low solar activity.

The distinctions between these two intervals were chosen to create a labelled data set of measurements with and without indication of ionospheric scintillation in the F-layer according to the characteristics observed in each measurement. The data set was characterised in terms of $S_4$, $\sigma_\phi$, intensity and (excess) phase power spectral density. Occultations pre-classified as moderate or strong scintillation (according to maximum $S_4$) were then visually inspected and labelled between two classes: "label 1"—disturbances with typical characteristics related to F-layer scintillation, and "label 0"—other disturbances.

The labelled data set was used to train an SVM algorithm with the goal to develop a model of automatic detection of ionospheric scintillation in low-latitude RO measurements. Different features were considered during the training of the classifier, including features exclusively related to the signal amplitude or phase and their combination. Furthermore, the selection of the most suitable kernel function of the SVM algorithm has also been evaluated.

Comparison between our results and a similar study not using RO measurements shows common outcomes and also distinctions [46]. In agreement, our results indicated that the $S_4$ index does not improve the performance of the SVM-model to correctly detect potential cases of ionospheric scintillation in low-latitude RO measurements. Our results also show the poor performance obtained when $\sigma_\phi$ was considered as the only training

feature. Regarding the kernel functions, the Gaussian kernel has not contributed to an increase in the performance of the classifier in comparison to the linear kernel. Finally, similar TPR were obtained in our study than when moderate and strong scintillations were used in the model training.

As a distinction, our model had to include information of the (excess) phase to achieve equivalent true positive alarm rates but at a higher rate of false positives. The different sorts of measurements considered in the SVM training, i.e., ground-based receiver [46], are likely related to the different performances observed. GNSS-RO measurements are significantly shorter, a fact that could have eventually influenced the labelling procedure in cases where a longer segment would be beneficial. The impact of this aspect should be assessed with the upcoming second generation of MetOp satellites (MetOp–SG), which will extend the plateau segment of the occultation towards higher SLTA points. Additionally, the current version of the data set of RO measurements is relatively small and it could be further extended and validated.

Nevertheless, the SVM-model described in this study sets a baseline for the use of machine learning algorithms in the detection of F-layer scintillation in RO measurements. The classifier has the potential to collect a large number of occultations with such characteristics and, consequently, to enable large-scale analysis of their influence on bending angle standard deviation, RIE and other atmospheric parameters over long time periods, for example, a solar cycle. In addition to MetOp, measurements from other RO missions may also be classified using the SVM-based model given the occultation geometry and data distribution (amplitude and excess phase) are equivalent. A detection model including classification of mid-, high-latitude and sporadic E-layer scintillation could also be assessed in future studies.

**Author Contributions:** Conceptualisation, V.L.-B.; methodology, V.L.-B., T.S.; software, V.L.-B.; validation, V.L.-B., T.S.; formal analysis, V.L.-B.; investigation, V.L.-B.; resources, M.I.P.; data curating, V.L.-B.; writing—original Draft preparation, V.L.-B.; writing—review and editing, V.L.-B, T.S., J.R., A.C., M.I.P., V.T.V.; visualisation, V.L.-B; supervision, M.I.P., V.T.V.; project administration, M.I.P., A.C.; funding acquisition, M.I.P., A.C. All authors have read and agreed to the published version of the manuscript.

**Funding:** This research was funded by Swedish National Space Agency (Rymdstyrelsen) under the National Space Technology Research Programme (NRFP), round 4.

**Data Availability Statement:** The RO data is provided by University Corporation for Atmospheric Research/COSMIC Data Analysis and Archive Center (UCAR/CDAAC) (http://cdaac-www.cosmic.ucar.edu/cdaac/products.html, accessed on 20 April 2021), the Kp index is archived by the Helmholtz Centre Potsdam—German Research Centre for Geosciences (GFZ) (https://www.gfz-potsdam.de/en/kp-index/, accessed on 20 April 2021), the solar flux index is provided by the Space Weather Canada—Solar Monitoring Program (https://spaceweather.gc.ca/solarflux/sx-en.php, accessed on 20 April 2021), the SYM-H index is provided by the World Data Center for Geomagnetism, Kyoto University (http://wdc.kugi.kyoto-u.ac.jp, accessed on 20 April 2021) and the IMF components are available at NASA/GSFC's Space Physics Data Facility—OMNIWeb's webpage (https://omniweb.gsfc.nasa.gov/, accessed on 20 April 2021). Files including the features used during the training/testing of the SVM algorithm are available at https://doi.org/10.5281/zenodo.4719592, accessed on 20 April 2021. Features are based on the data given in *atmPhs* and *atmPrf* files provided by UCAR/CDAAC.

**Acknowledgments:** The author would like to thank the anonymous reviewers for their contribution to improve the quality of the manuscript. In addition, Ana Luiza Dallora Moraes (Blekinge Institute of Technology, Sweden) and Alexandre Becker Campos (Aeronautics Institute of Technology—ITA, Brazil) for discussions on the development and validation of the SVM model.

**Conflicts of Interest:** The authors declare no conflict of interest. The funder had no role in the design of the study; in the collection, analyses, or interpretation of the data; in the writing of the manuscript, or in the decision to publish the results.

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
