# Peer review of "Supervised Detection of Ionospheric Scintillation in Low-Latitude Radio Occultation Measurements"

_remotesensing, doi:10.3390/rs13091690_

Round 1

Reviewer 1 Report

REVOEWER’S REPORT

TITLE: Supervised Detection of Ionospheric Scintillation in Low-Latitude Radio Occultation Measurement     

It is a really nice study with an aim of detecting occultation ionospheric scintillation features utilizing the low latitude radio occultation measurements using the machine learning technique in particular Support Vector Machine (SVM) algorithm. Though the idea and the paper buildup seems sound but still there are significant gaps in the paper which authors should omit in order to turn their paper in a significant contribution to the space weather community. The comments and advice to the paper are as following.

 “123 performance in terms of true and false positive rates (TPR, FPR) given in the Receiver Operating Characteristic (ROC) curve. “

Author should define very clearly and briefly TPR and FPR.

“124 The main goal of the model is to automatically detect occultations containing typical spectral features related to ionospheric scintillation in F-layer, as described in the scintillation theory. “

Merely saying as in scintillation theory is not appropriate, it has to be listed exactly what spectral features and in what extent and with appropriate references.

“126 The classifier has the potential to collect cases with such characteristics and to enable large-scale analysis of their influence on bending angle standard daviation, RIE and other atmospheric parameters over long time periods, e.g., a whole solar cycle.”

This paragraph is not appropriate for the introduction section; it may be suitable for abstract, methodology or summary and conclusion part. As the author are introducing and building up background of their paper and the strong statements there are making in this paragraph for which they haven’t presented any evidences so far.

“141 The effects of the ionospheric refractivity, including regions of irregularities, are mostly observed within 30 km to 100 km tangent altitude in RO measurements. “

Please include the references here to support your statement.

“147 In order to automatically limit the segment of the occultation, a truncation threshold was set to 90%of the signal-to-noise (SNR) mean. “

On what basis this 90 % SNR mean threshold was decided. Here we got a little problem. Either authors support their assumption by adding some references and discussing it well or add some figures and text to explain it well.

“ 148 Then, low- and high-dimensional featureswere computed from……”

What are low and high dimensional features?? I struggled to get it throughout the paper.

“150 i.e. S4 (low-dimensional) and intensity PSD (high-dimensional)..”

Please remember amplitude scintillation index is fourth moment of the received signal amplitude and it is a lower index, please correct it throughout the paper and figures wherever it appears.

“175 Therefore, indexes alone can leave some..”

Do you mean Indices???

Figure 1 description is more or less is just naming the caption for each figures. However there are several significant information in the detrended phase as well as amplitude scintillation index power spectrucms. Please extend this section and explain them by including how the SNR changes are reflecting into PSD.

223-228 authors do not discuss how they combined RO observation with the ground based measurements. First thing they should discuss what RO satellites or observation is in close proximity of the ground based observation. How do they vary with each other?  There must be a criterion to combine them and it has to be clearly discussed this section. 

Figure 2 Please include sym-H and IMF in this figures as well. Kp is a three hourly index authors should not rely on only this index.

Figure 5-8 Bar does not seems good choice for plotting the occurrence rate why not simply include solid or dotted nan lines to show it ??? Bars may hide some information.

Figure 9-10

If p < 3, an outer scale is present, the limiting value of the scintillation index is unity. The approach to this limit may be monotonic, or a local maximum may be achieved with S4 > 1 due to strong focusing.

However, the limiting correlation length is dictated by the high‐frequency portion of the phase spectrum when it is shallow, as might be expected.

These figures are not well discussed, please extend this portion and use equations to explain it better.

Level 0 data, is perhaps less calibrated or not calibrated data, authors didn’t mentioned purpose of using this.

Figure 11; please refer to my previous comment for using bars.

Moreover I don’t think combining label 0 and label 1 observation is a smart idea, rather comparing them may be more helpful in all the Figures that used this idea.

Why not plot the occurrence of label 0 and labels 1 using two different color line curves; and then discuss them. It will certainly make more sense than what has been presented here.

This ambiguity here make me think to return back this paper to the author, but considering the other merits of the paper I am still positive and thinking that these issues can be resolved during the review process.

 Figure 14-17

It is interesting what have been presented, however comparison of RMSE for the different modeled combination will justify what have been presented so far. Please include RMSE comparison in these figures or separately by explicitly mentioning the name of combinations.

Conclusion part has to be rewritten following the addition in the RMSE plots in the figures, I am sure it will change entirely the way it is presented.

Author Response

Thank you for your comments in our manuscript. Please see the attachment containing our replies to each one of them. Mentions to specific lines refer to the document containing tracked changes in the manuscript.

Reviewer 2 Report

good job

Author Response

Thank you for your comments in our manuscript.

Reviewer 3 Report

The main contribution of this submission is the SVM-based classifier for analysis of the ionospheric scintillation in low-latitude radio occultation measurements. The submitted manuscript is well written and suitable for the journal.

Comments:

  1. the brief introduction to machine learning (ML) should be included in the state of the art description; Please clearly separate between regression and classification to help readers not familiar with ML to follow-up.
  2. Is is not clear why SVM was selected. 
  3. There are many results these days derived for reliable ionospheric total electron content nowcasting using GNSS. authors may wish to analyse the background and include more relevant references.
  4. Authors are supposed to cite original papers/book where SVM (Vapnik-Chervonenkis theory) was proposed.

Author Response

(The authors gave the same response as above.)

Round 2

Reviewer 3 Report

The revised version can be accepted.